# The Impact of Human Genetic Polymorphisms on Rotavirus Susceptibility, Epidemiology, and Vaccine Take

**DOI:** 10.3390/v12030324

**Published:** 2020-03-17

**Authors:** Sumit Sharma, Marie Hagbom, Lennart Svensson, Johan Nordgren

**Affiliations:** 1Division of Molecular Medicine and Virology, Department of Clinical and Biomedical Sciences, Linköping University, 58183 Linköping, Sweden; sumit.sharma@liu.se (S.S.); marie.hagbom@liu.se (M.H.); Lennart.t.svensson@liu.se (L.S.); 2Division of Infectious Diseases, Department of Medicine, Karolinska Institute, 171 76 Stockholm, Sweden

**Keywords:** rotavirus, histo-blood group antigens, susceptibility, rotarix, RotaTeq, disease burden, vaccine take, epidemiology

## Abstract

Innate resistance to viral infections can be attributed to mutations in genes involved in the immune response, or to the receptor/ligand. A remarkable example of the latter is the recently described Mendelian trait resistance to clinically important and globally predominating genotypes of rotavirus, the most common agent of severe dehydrating gastroenteritis in children worldwide. This resistance appears to be rotavirus genotype-dependent and is mainly mediated by histo-blood group antigens (HBGAs), which function as a receptor or attachment factors on gut epithelial surfaces. HBGA synthesis is mediated by fucosyltransferases and glycosyltransferases under the genetic control of the *FUT2* (secretor), *FUT3* (Lewis), and *ABO (H)* genes on chromosome 19. Significant genotypic and phenotypic diversity of HBGA expression exists between different human populations. This genetic diversity has an effect on genotype-specific susceptibility, molecular epidemiology, and vaccine take. Here, we will discuss studies on genetic susceptibility to rotavirus infection and place them in the context of population susceptibility, rotavirus epidemiology, vaccine take, and public health impact.

## 1. Introduction

Infectious diseases have been a major driver of the evolution of the human genome via the selection of alleles associated with infection and pathogenesis. The recent decades have seen a large increase in understanding of the alleles affecting susceptibility and severity for several pathogens, including viruses, bacteria, and parasites. Rotavirus, the most common agent of severely dehydrating gastroenteritis in children worldwide, is highly contagious and transmittable in the pediatric population. Most children would have had several rotavirus infections by five years of age. In spite of this high infectivity and rapid transmission, recent observational studies have shown that a subpopulation appears resistant to infection and clinical disease of particular rotavirus genotypes. This has been shown to be mediated through expression of human histo-blood group antigens (HBGAs), mainly the types controlled by the *FUT2* (secretor), *FUT3* (Lewis), and *ABO* genes. Both in vivo and in vitro studies have shown that this resistance is dependent on the rotavirus genotype, and in some cases perhaps also between different rotavirus strains of the same genotype.

Furthermore, the two globally licensed live attenuated rotavirus vaccines, Rotarix and RotaTeq, have been associated with similar susceptibility factors as natural infections. As HBGA distribution varies widely between populations and ethnic groups, this is an important factor to consider regarding vaccine efficacy and protection in different populations.

Here, we will review previous and recent studies on rotavirus infections in relation to host genetic susceptibility. These findings will be discussed in the light of rotavirus epidemiology, population susceptibility, zoonotic transmission, and rotavirus vaccination.

## 2. Rotavirus Classification and Genotypes

Rotaviruses belong to the family *Reoviridae*, with a naked icosahedral triple-layered capsid containing 11 double-stranded (ds) RNA gene segments. These encode for six structural proteins (VP) and five to six nonstructural proteins (NSP). Ten groups (A–J), or species, have been described based on serological or genetic diversity of the middle layer VP6 protein [1]. Group A rotavirus is by far the most commonly found in humans and will be referred to as rotavirus henceforth in this review. The outer capsid glycoprotein VP7 and protease-activated spike protein VP4 elicit neutralization antibodies and are used to define glycoprotein (G) and protease sensitive (P) serotypes or genotypes, respectively. Many different G and P genotypes have been identified, but only a few are commonly observed in humans: G1–G4, G9, G12 and P[4], P[6], and P[8] [2,3].

There also exists a complete nucleotide sequence-based genome classification for rotavirus strains, assigning each genome segment a specific genotype. This classification is used for better understanding of rotavirus evolution, reassortment, and zoonotic transmission [3].

The VP4 protein (P genotype), which is post-translationally cleaved into VP8* (glycan-binding domain) and VP5* polypeptides, is responsible for cellular attachment and entry as well as HBGA binding in vitro [4]. The P genotypes thus determine the pattern of genetic susceptibility and will therefore be the focus of this review.

## 3. In Vitro Binding Studies Show Rotavirus Binding to HBGAs in a P Genotype–Dependent Manner

Earlier studies showed animal rotaviruses to recognize terminal sialic acids, but subsequent studies have shown that some animal rotaviruses and almost all human rotaviruses are sialidase-insensitive, with some strains recognizing internal sialic acid [5]. Direct experimental evidence that HBGAs are important for rotavirus infections first came from in vitro binding studies published in 2012 using both synthetic VP8* particles and triple-layered rotavirus virions. These studies demonstrated binding to carbohydrate conjugates as well as to HBGA characterized saliva [6,7] and further identified type 1 HBGAs, including secretor, Lewis, and ABO antigens as important attachment factors for human rotavirus.

A subsequent study [8] of a neonatal rotavirus strain of P genotype P[11] reported binding to type II precursor glycans. Core glycan synthesis is constitutive in most cell types, and their modification is believed to be developmentally regulated, thereby providing a plausible mechanism for explaining why this rotavirus genotype is more common in neonates [8] (see more in Section 9: Neonatal Infections and Host Genetics).

More recently, a study using VP8* constructs from clinically isolated P[4], P[6], and P[8] genotypes reported that P[4] and P[8] bind to secretor- and Lewis-positive saliva, and less to blood group B, whereas genotype P[6] rotaviruses bound to Lewis-negative saliva independent of secretor status [9]. These in vitro results corresponded well with the epidemiological data from studies in which these strains were first detected [9].

Moreover, the VP8* construct of the P[8] genotypes within the rotavirus vaccines Rotarix and RotaTeq also has binding affinities to secretor- and/or Lewis b-positive saliva, and therefore is similar to the binding pattern of other wild-type P[8] strains [9,10]. A recent study [11] however, demonstrated a variation in the ligand affinities among different P[8] lineage through subtle structural differences, and that Rotarix differs from other P[8] strains of the same lineage I, in showing a lack of interaction with the H type 1 antigen. These results suggest that different P[8] strains, including Rotarix, may also have a different host genetic profile, as indicated by some epidemiological studies (Table 1).

To summarize, in vitro binding studies have reported that the globally predominant P[4], P[6], and P[8] genotypes as well as other less common P genotypes infecting humans recognize different HBGAs. The specific binding affinities correspond well with epidemiological observations, as will be described subsequently in this review.

## 4. The HBGA Biosynthesis Pathway

HBGAs are synthesized by the sequential addition of monosaccharides onto a precursor, which is catalyzed by glycosyltransferases encoded by the *ABO (H)*, secretor (*FUT2*), and Lewis (*FUT3*) genes. The FUT2 enzyme, which forms the H antigen by addition of a fucose to the precursor, is active mostly in epithelial tissues. HBGAs under the control of FUT2 are therefore present mainly on the mucosal epithelia of the respiratory, genitourinary, and digestive tracts and as free oligosaccharides in body fluids such as saliva. The FUT3 enzyme converts the H type 1 antigen to Lewis b through the addition of a fucose. Moreover, the A or B enzymes encoded by the *ABO* gene can add either an acetylgalactosamine or a galactose to the H antigen. Individuals with a non-functional FUT2 enzyme are termed non-secretors, given the absence of ABO (H) groups in the saliva and mucosa. These individuals express Lewis a if they have a functional FUT3 enzyme that catalyzes the addition of a fucose residue to the H type 1 precursor. Homozygotic inactive *FUT3* gene carriers lack Lewis a and b structures and are termed Lewis-negative [9,22].

## 5. Rotavirus Susceptibility In Vivo Is Strongly Associated with HBGAs in a P Genotype–Dependent Manner

Following the first in vitro binding studies, several observational studies have investigated the association between different HBGA phenotypes and/or genotypes and susceptibility to rotavirus infection in vivo (Table 1). First, a study from France found that rotavirus P[8] infections were completely absent in individuals with *FUT2* homozygous nonsense mutation, yielding the non-secretor phenotype [20]. Subsequently, a study from Burkina Faso reported that P[8] and P[4] genotypes infected only secretor- and Lewis-positive children (Lewis b phenotype), whereas P[6] rotavirus predominantly infected children with the Lewis-negative phenotype independent of secretor status [12]. Several subsequent studies from several countries and continents (Table 1) reported that positive secretor status was strongly associated with susceptibility to the P[8] and P[4] genotypes. Subsequent studies also verified a strong association between P[6] susceptibility and the Lewis-negative phenotype, independently of secretor status (Table 1).

Some discrepancies have been found between studies, mostly regarding the P[8] genotype, which most studies have investigated. While most studies have reported a strong association between positive secretor status and susceptibility, some studies have reported Lewis positivity, independent of secretor status, as a susceptibility factor [16,21], while others have reported that secretor- and Lewis-positive status (Lewis b phenotype) may be a stronger susceptibility marker than only secretor-positive status [12,15]. Although fewer studies on P[4] are available, similar discrepancies have been reported, with some studies showing that secretor and Lewis positivity (Lewis b phenotype) are markers of susceptibility rather than only secretor positivity [15]. One study [16], reported no P[4] infections in non-secretors, but Lewis-positive non-secretors were susceptible to P[8] infections. The reduced estimate of vaccine efficacy in this study was thus mediated by the complete protection of non-secretors to P[4], and not P[8], infections.

To summarize, observational studies have provided strong evidence that secretor and Lewis antigens are important for susceptibility to rotavirus in a P genotype–dependent manner. Positive secretor status is strongly associated with P[8] and P[4] infections, but the discrepancies observed between studies warrant more investigation. The putative reasons include a strain-dependent susceptibility, methodological differences between studies, differences between mild and severe rotavirus cases [15,16], or the lack of sufficient samples for reliable statistical analysis. All studies on the P[6] genotype to date have reported a strong association with Lewis negativity independent of secretor status. A few studies have also associated ABO blood group with susceptibility [19], but more studies are warranted.

## 6. Secretor-Positive Adults Have Significantly Higher Immunoglobulin G (IgG), IgA, and Neutralization Antibody Titers to Rotavirus Compared to Non-Secretors

Several studies have investigated rotavirus-specific antibody titers in adults in association with HBGAs. Higher anti-rotavirus antibody titers would likely reflect a larger number of previous infections, making it an indirect marker of susceptibility. A study from Sweden [23] found that secretors had higher serum rotavirus IgG titers as well as higher neutralization antibody titers to a P[8] strain, but not to a P[6] strain, likely reflecting that P[6] infections are rare in Sweden (Table 2). A subsequent study from France [9] also reported higher neutralization antibody titers to P[8] in secretors compared to non-secretors. A study from China [13] also reported higher serum rotavirus IgG titers in secretors. Further, a study from Spain measuring salivary rotavirus IgA titers [24] reported higher antibody titers in secretors compared to non-secretors (Table 2). The effect of Lewis and ABO antigens have been investigated by Gunaydin et al. [23], and Zhang et al. investigated the effect of ABO antigens [13]. No association between Lewis or ABO status to rotavirus antibody titers was reported.

To summarize, serological studies have reported higher rotavirus-specific IgG titers in serum and IgA titers in saliva in secretors as compared to non-secretors. Moreover, secretors have higher neutralization antibody titers to P[8] and P[4] compared to non-secretors. No effect regarding ABO or Lewis status has been observed, but due to the limited numbers more studies are warranted.

## 7. Population Genetics and Epidemiology of Rotaviruses

The fact that populations vary considerably in relation to HBGA genotype and phenotype raises the question of whether this will affect rotavirus epidemiology at population level. For example, approximately 75–80% of European, North American, Central Asian, and several African populations are secretors [22,25], whereas in Mesoamerica, the prevalence of secretor positives can reach ~90–95% [18,26]. The Lewis-negative phenotype, moreover, is significantly more prevalent in several African countries or ethnicities (20–35%) compared to European, North American, and some Asian populations (6–11%) [25] (Figure 1).

Globally, three P genotypes, P[4], P[6], and P[8] are common in humans, with others being detected occasionally. A major difference in global rotavirus epidemiology related to P genotypes is P[6], which is common in sub-Saharan Africa, and to some extent in Southeast Asia, but is almost completely absent elsewhere [2,29]. As reviewed here, there is strong epidemiological evidence that the Lewis-negative phenotype is associated with susceptibility to genotype P[6]. Therefore, as the Lewis-negative phenotype is much more prevalent in sub-Saharan Africa, this is likely the reason why P[6] genotype is more common there. In turn, the P[4] and P[8] genotypes, which are associated with globally common HBGA phenotypes such as secretor and Lewis b, account for most of the infections in humans. The higher diversity and more similar relative proportions of different HBGA phenotypes in sub-Saharan Africa likely also render the P genotype distribution more diverse in these countries.

The available body of evidence therefore strongly suggests that population genetics shapes both global and regional rotavirus epidemiology through differential expression of HBGAs. No clear consistent differences in clinical severity between the globally dominant P genotypes has been observed to our knowledge, and it is unclear how and in which way the rotavirus disease burden would be affected. As non-secretors are less susceptible to the globally common P genotypes P[4] and P[8], countries with a high proportion of non-secretors would accordingly have less rotavirus disease caused by these genotypes. Some studies have indeed suggested that rotavirus burden is dependent on ethnicity [18,30], although it is difficult to rule out other factors such as socioeconomics.

In sub-Saharan Africa, however, the relatively high Lewis-negative proportion of non-secretors would be susceptible to P[6] (Figure 2), which implicates a larger overall population susceptibility for rotavirus disease. More studies are warranted to understand if and how HBGA phenotype distribution affects rotavirus disease burden at population level.

## 8. Zoonosis and Host Genetics

An interesting aspect of HBGAs and rotavirus susceptibility is zoonotic infections. Similar human HBGAs (ABO, H, and Lewis families) are also observed in many animals, leading to shared HBGA antigens between humans and some animals [4]. These shared HBGAs may be responsible for rotavirus cross-species transmission [4,31].

Only a few P genotypes infecting animals are occasionally observed in humans, with the particular exception of the P[6] genotype. Rotavirus strains of P[6] genotype commonly infect pigs [32], and it is the only P genotype that is common in both animals and humans. In humans, there is a strong association between the Lewis-negative phenotype and genotype P[6]. Although not widely investigated, some studies have suggested that the Lewis-negative phenotype is common in pigs [33]. A study from Nicaragua, which used human monoclonal antibodies, reported that the saliva of the majority (~60%) of pigs was Lewis-negative, with both Lewis a and Lewis b antigens detected in some pigs [33]. Other studies have reported a predominance of H type 1 glycans in pig gastrointestinal tract, whereas Lewis antigens were present in some pigs but lacking in others [34,35]. Studies on rotavirus P genotype susceptibility in pigs of different breeds and HBGA expressions would be important to shed more light on this subject.

Other studies with a P[19] genotype, which is common in pigs and occasionally observed in humans, have shown binding to mucin core 2 and type 1 HBGA [36,37,38]. Of further interest is the P[14] genotype, which is common in several animals species including cattle [39] and occasionally observed in humans. It recognizes the A antigen, a HBGA common in humans [6]. Perhaps zoonotic transmission of this genotype only occurs when secretor positive type A humans come into close contact with P[14]-infected animals, which is partially supported by in vivo data [6,40]. Although the prevalence of blood type A varies widely between populations, the relatively overall low proportions of secretor and type A humans may be one of the reasons why human-to-human transmission of P[14] genotype is rare. A similar association for susceptibility of the A antigen has also been reported for P genotypes P[9] and P[25] [4,41].

To summarize, the genetic aspects of zoonotic rotavirus transmission warrants more studies. The available evidence suggests that the relatively uncommon P genotypes P[9], P[14], and P[25] are transmitted from some animals (e.g., pigs) to A antigen positive humans. The common P[6] genotype might be transmitted through H type 1 and/or precursor glycans, made available through the Lewis-negative phenotype, common in both pigs and some human populations, particularly in sub-Saharan Africa.

## 9. Neonatal Infections and Host Genetics

Rotavirus infections are relatively common in neonates and are usually asymptomatic [42]. In many geographical settings, the rotavirus genotypes infecting neonates are usually different from those that infect older children [43,44,45,46]. Two common neonatal P genotypes are P[6] and P[11]. In vitro, P[11] recognizes poly-LacNAc (poly-*N*-acetyllactosamine), a type 2 HBGA precursor glycan, whereas P[6] recognizes both H type 1 and its precursor [8,47]. The expression of these precursor molecules is believed to be developmentally regulated, which could explain the age limit of rotavirus infection and disease for some specific genotypes [8,48].

While genotype P[11] is rare in older infants, the P[6] genotype is common in sub-Saharan Africa, with susceptibility being strongly associated with the Lewis-negative phenotype. A recent study has suggested that small differences in amino acid distribution between VP8* of P[6] strains isolated from neonates compared to P[6] infecting older children restrict their ability to bind branched glycans [47]. Unbranched glycans, including unbranched type 1 precursor glycans, are more abundant in the neonatal gut [47,49]. Accordingly, both neonatal strains P[11] and P[6] likely recognize precursors common in the neonatal gut. The presence of P[6] in older children might be associated with these small amino acid differences or to a greater availability of precursor molecules in Lewis-negative children.

In summary, some evidence suggests that the age restriction of some P genotypes infecting neonates is related to the developmental regulation of HBGAs. The observed amino acid differences between P[6] strains infecting neonates compared to that infecting older children warrants more investigation.

## 10. Rotavirus Vaccine Take Is Associated with HBGAs

Live oral attenuated rotavirus vaccines have successfully reduced the mortality and morbidity of rotavirus disease worldwide [50]. However, the efficacy varies considerably between regions, with low-income countries, particularly sub-Saharan Africa and Asia, demonstrating lower vaccine efficacy [50]. Several reasons may account for this lower efficacy, including early rotavirus infections, transplacental antibodies, concomitant infections, enteric dysfunction, nutritional status, and gut microbiota [51,52].

As both Rotarix and RotaTeq are live attenuated vaccines and contain the P[8] genotype, it has been hypothesized that children’s susceptibility to the vaccine strains would vary as that for natural infections of the same genotype. In a scenario wherein no infection and subsequent replication occurs in the vaccinated child, the immune response is likely to be suboptimal.

Studies from Nicaragua [27,53], Pakistan [54], Ghana [55], and Malawi [15] have tested this hypothesis by measuring IgA seroconversion and/or vaccine strain shedding in association to HBGAs (Table 3). Five studies investigated Rotarix whereas one study investigated the RotaTeq vaccine. All studies reported a lower proportion of seroconversion in non-secretors as compared to secretors, albeit to different degrees (Table 3). An important factor to consider here is that many studies investigated seroconversion after two to three doses, therefore there is a higher likelihood of early natural infections that may affect interpretation of the results. No studies found any effect of Lewis antigens, apart from the studies in Nicaragua showing a lower proportion of seroconversion in children with the Lewis a phenotype after one dose, which might reflect the premise that all Lewis a carriers are also non-secretors. The number of Lewis-negative non-secretors was too low to make a statistical comparison. The same study also assessed RotaTeq and found the same results, i.e., no seroconversion for Lewis a phenotype children, but the numbers were too low for reliable interpretation [53].

Studies from Malawi [15] and Nicaragua [27] have also measured vaccine shedding after vaccination, as determined ≥4 days post-vaccination, to measure active vaccine strain replication. After one dose of Rotarix, no shedding was observed in non-secretor infants in Nicaragua; while in Malawi, non-secretors had a significantly lower proportion of shedding. In Malawi, however, no difference in shedding proportion between secretors and non-secretors were observed after the second dose. The study in Nicaragua further investigated shedding of RotaTeq and observed no shedding in children with the Lewis a phenotype. A recent study investigated the neonatal RV3-BB vaccine (G3P[6] genotype) [56] and found a similar cumulative vaccine take irrespective of secretor status. Cumulative vaccine take was defined as seroconversion and/or vaccine strain shedding after any of the three doses given, which might affect interpretation of the results.

Moreover, some of these studies investigated the effect of ABO, and reported different results. While some studies showed that ABO blood group affected vaccine take, others observed no such association (Table 3).

To conclude, a relatively large body of evidence shows that the secretor-positive phenotype is associated with higher vaccine take, as measured by seroconversion and/or vaccine strain shedding, for Rotarix. No strong effects of the Lewis phenotype have been observed. More studies on the effect of ABO, as well as studies on RotaTeq and newly developed vaccines such as RV3-BB are warranted. It is also important to take into account methodological differences between studies, as well as definitions of secretor and Lewis status. Most studies use saliva-based ELISA assays, often with different antibodies, whereas some studies use *FUT2/FUT3* genotyping, with some studies using both approaches to different degrees. As definitions, particularly on phenotype level, of secretor and Lewis status are not universally standardized, this may account for some of the differences between studies. Furthermore, secretor and Lewis positivity is not absolute in that not all type one chain precursors are always entirely converted into H or Lewis antigens by the FUT2 and FUT3 enzymes, respectively. Core structures may thus coexist together with H and Lewis antigens [12,57].

## 11. Will Reduced Vaccine Take Translate to More Clinical Vaccine Failures?

Studies thus suggest that HBGAs, particularly secretor status, affect vaccine take, as measured by both IgA seroconversion and viral vaccine shedding. The question remains whether this will translate to a higher degree of vaccine failures, as children resistant to the live vaccines would also be resistant to naturally circulating rotavirus genotypes of the same P genotype. As both P[4] and P[8] genotypes recognize and infect children of similar HBGA phenotypes, a major determinant is likely to be the proportion of P[6] circulating in a given population.

Pollock et al. [15] observed that non-secretor children in Malawi had reduced risk of rotavirus gastroenteritis after vaccination because they were naturally resistant to the majority of the naturally occurring P genotypes. A study in Bangladesh [16] reported a similar result. Bangladesh had low numbers of P[6] infections, while that in Malawi was relatively high (20%). Furthermore, a study from the USA [18] on a highly vaccinated population reported that non-secretors were protected against severe rotavirus gastroenteritis, with most infections being due to P[8] genotypes.

As non-secretors appear less susceptible to the live vaccines, and also more resistant to P[4] and P[8] infections, the susceptible pool after vaccination is reduced. An interesting subgroup of the population in this aspect would be children that are both non-secretors and Lewis-negative. This subgroup is, however, rare. In sub-Saharan Africa, where it is more common, it is only present in around 7% of the population [12] (Figure 2). However, it is virtually non-existent in Europe and North America (Figure 2). We may therefore speculate that HBGA-related vaccine failures are associated with the proportion of secretor- and Lewis-negative individuals, as they are resistant to the live vaccines but are susceptible to infections of the P[6] genotype. As the prevalence of this subgroup is low, large studies in sub-Saharan Africa are likely needed to test the hypothesis.

To conclude, more studies from different regions and countries are needed to assess whether, and to what extent, population distribution of HBGAs account for more vaccine failures. The important factors to consider are likely both the proportion of non-secretors and Lewis-negative individuals as well as the prevalence of P[6] rotavirus infections.

## 12. Human Intestinal Enteroids: A Novel Model for Studying Genetic Susceptibility to Rotavirus Infections

Although different cell lines support rotavirus growth, several studies have suggested that they are not a good model for susceptibility studies involving HBGA, as in vitro infection of transformed cell lines is independent of HBGA expression [9].

Recently, human intestinal enteroids (HIEs), derived from human small intestinal tissue, were used successfully to support human norovirus replication [58,59]. In such studies, there is concordance between the genetic susceptibility of different HIEs to norovirus and in vivo epidemiological data [22]. Accordingly, HIE might be a suitable model for investigating HBGA-mediated rotavirus susceptibility as well.

A study performed with rotavirus on HIE [60] found no major differences for regarding rotavirus strain Ito (genotype G3P[8]) and infection of HIEs of different secretor status. They further observed that the Rotarix stain (G1P[8]) did not grow well in one of three HIEs genotyped as secretor-negative [60]. Our own experiments have shown that HIEs isolated from a secretor-positive individual were more susceptible to infection with the F45 strain (G9P[8] genotype) and a clinical rotavirus isolate of genotype G3P[8] (Figure 3), as compared with HIEs of a non-secretor individual.

To conclude, HIEs are a promising tool for gaining more insight into host restriction factors regarding rotavirus infections. Future studies with different rotavirus genotypes and strains using HIEs of different secretor, Lewis, and ABO profiles are warranted to answer many outstanding questions on genetic susceptibility and rotavirus genotype-specific infection.

## 13. Conclusions and Outstanding Questions

Studies performed during the last decade have shed light on many aspects of genetic susceptibility to rotavirus infections. The three predominant human genotypes, i.e., P[4], P[6], and P[8], have all been associated with specific HBGAs. The differences in HBGA specificity between genotypes explains their relative prevalence globally and in different populations, e.g., the relatively high prevalence of P[6] in sub-Saharan Africa.

Moreover, the importance of HBGA in neonatal as well as zoonotic infections is intriguing. Neonatal strains bind to precursor and unbranched glycans that are more prevalent in the neonatal gut, which can explain the age restriction of some rotavirus genotypes. Similarly, it has been suggested that the HBGA moieties which are shared/similar between animals and human are a determining factor for zoonotic transmissions. Clearly, more studies are needed in this regard to provide more in vivo data to understand when and how zoonotic transmission occurs.

Studies on vaccine take have generally found that non-secretors have lower vaccine take than secretors. However, the interplay between HBGAs, circulating P genotypes, and the potential differences between mild and severe rotavirus infections warrants more studies to establish the effect on vaccine efficacy and protection at population level. It is important to consider that individuals resistant to the live vaccines will also be resistant or less susceptible to the wild-type strains. Indeed, studies from a few countries have shown that non-secretors, although less immunized with the vaccines, are at reduced risk of subsequent rotavirus disease. More studies from countries with differences particularly regarding the Lewis-negative phenotype and circulating P[6] strains will be important for addressing this.

Different in vivo and in vitro studies have reported small but significant differences in the susceptibility and binding patterns between HBGAs for the same P genotypes. The reasons for these discrepancies are unclear, but they might be associated with subtle differences between strains of the same genotype. As proposed [15,16], it is also possible that non-secretor status may limit the severity of P[8] rotavirus, but may be permissive to milder infection. As most epidemiological evidence so far has been from studies on secretor status and severe diarrhea requiring hospitalization, more studies regarding this are needed. The HIE, which mimics the human intestinal epithelium, will be a valuable tool for addressing some of these outstanding questions and for other future mechanistic studies on genotype-specific susceptibility.

## Figures and Tables

**Figure 1 viruses-12-00324-f001:**
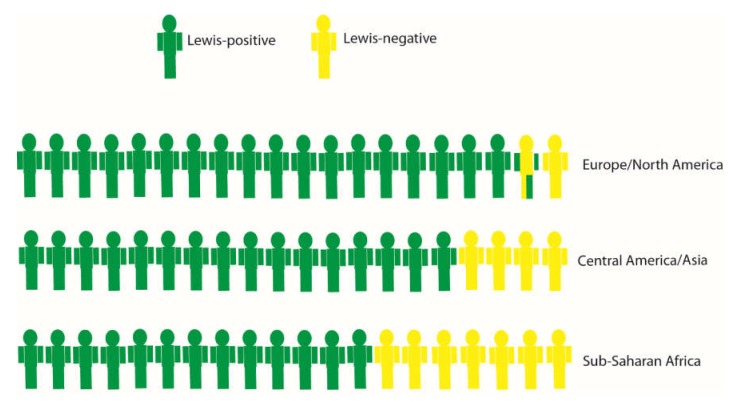
A simplified rough estimation of Lewis status in different geographical settings (~7.5%, 20%, and 33% in Europe/North America, Central America/Asia, and Sub-Saharan Africa, respectively). The proportion of Lewis-negatives vary widely between populations [12,15,16,23,26,27,28]. Lewis-negative individuals are susceptible to P[6] rotavirus infections, and the relative proportion of P[6] rotavirus infections in a population correspond reasonably well to the proportion of Lewis-negatives in that population. Rotavirus P[6] infections are virtually nonexistent in Europe and North America, but common in Sub-Saharan Africa.

**Figure 2 viruses-12-00324-f002:**
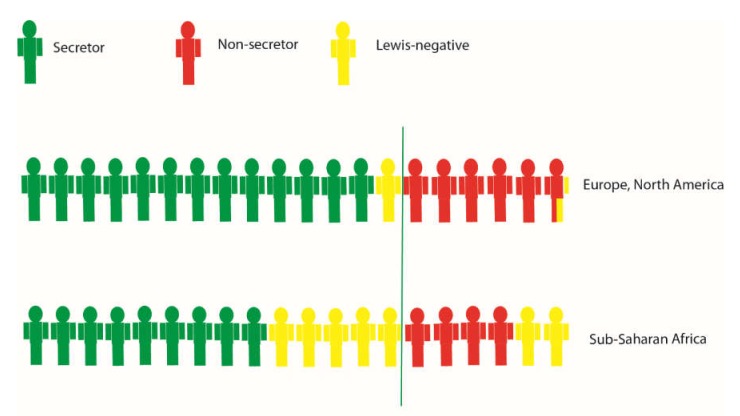
A simplified rough estimation of secretor and Lewis status in different geographical settings (~20% non-secretors and ~7.5% Lewis-negative in Europe and North America and 20% non-secretors and ~33% Lewis-negative in Sub-Saharan Africa) [12,15,16,22,23,26,27,28]. Studies have suggested that non-secretors have a lower vaccine response but would also be protected against circulating wildtype strains of the same genotypes which predominantly infect secretors. However, the Lewis negative proportion of non-secretors, could potentially be susceptible to P[6] strains even after vaccination.

**Figure 3 viruses-12-00324-f003:**
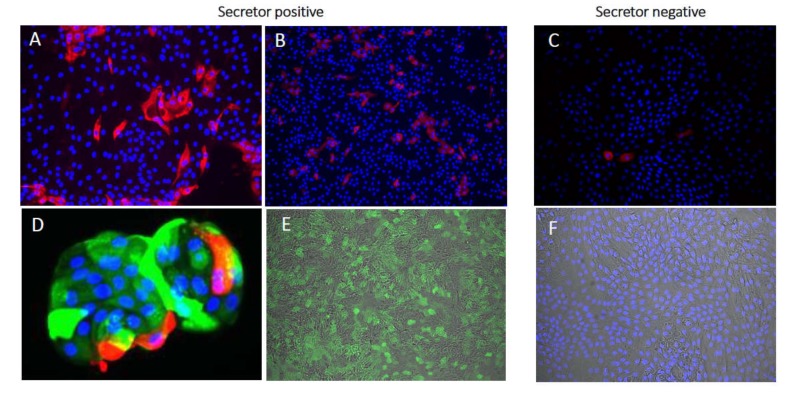
Human intestinal enteroids (HIEs) infected with a clinical human rotavirus isolate of genotype G3P[8]. Monolayers of differentiated HIEs from jejunum of secretor-positive (HIE 004) and secretor-negative (HIE 002) individuals. Immunofluorescence staining was performed at 24 h post infection. Rotavirus infected secretor-positive (**A**, 20× **B**, 10×) and to a lesser extent secretor-negative (**C**, 10×) monolayers. (**D**, ~100×) secretor-positive HIE stained for α1,2 fucose and rotavirus. Monolayer of secretor-positive (**E**, 20×) and secretor-negative (**F**, 20×) showing overlay of bright-field and α1,2 fucose UEA-I FITC staining. Red = rotavirus, Alexa594; Green = α1,2 fucose, FITC; and Blue = nuclear staining, DAPI.

**Table 1 viruses-12-00324-t001:** Association between histo-blood group antigens and susceptibility to rotavirus stratified by P genotype. Included are studies that reported P genotypes with at least 5 samples.

P-Genotype	Country	Lewis	ABO	Secretor	Reference
**P[4]**					
	Nicaragua	Only Lewis b infected	Not investigated	Only secretors infected	[12]
	China	Lewis a less infected	No association	Secretors more infected	[13]
	China	No association	No association	Only secretors/partial secretors infected	[14]
	Malawi	Lewis negative less susceptible	No association	Secretors more infected	[15]
	Bangladesh	Only Lewis b infected	Not investigated	Only secretors infected	[16]
	Vietnam	Lewis b present in all infected	No association	Only secretors/partial secretors infected	[17]
	USA	Not investigated	Not investigated	Only secretors infected	[18]
**P[6]**					
	Burkina Faso	Lewis negative more susceptible	No association	No association	[12]
	Malawi	Lewis negative more susceptible	No association	No association	[15]
	Bangladesh	Lewis negative more susceptible	Not investigated	No association	[16]
	Vietnam	Lewis negative more susceptible ^a^	No association	Not investigated	[17]
**P[8]**					
	Burkina Faso	Only Lewis b infected	No association	Only secretors infected	[12]
	Nicaragua	Only Lewis b infected	Not investigated	Only secretors infected	[12]
	China	Lewis a less infected	No association	Secretors more infected	[13]
	China	Lewis a less infected	No association	Secretors more infected	[14]
	Spain	Lewis b more infected	Blood group A and AB more infected compared to O	Secretors more infected	[19]
	Malawi	Lewis negative less susceptible	No association	Secretors more infected	[15]
	Bangladesh	Lewis negative less susceptible	Not investigated	No association	[16]
	Vietnam	No Lewis a infected	No association	Only secretors/partial secretors infected	[17]
	France	Not investigated	Not investigated	Only secretors infected	[20]
	USA	Not investigated	Not investigated	Apart from 1, only secretors infected	[18]
	Tunisia	Only Lewis positives infected	No association	No association	[21]

^a^ based on low OD values for Lewis a and Lewis b.

**Table 2 viruses-12-00324-t002:** Serological markers in association with histo-blood group antigens.

Country	Type of Response	ABO	Lewis	Secretor Status	Reference
Sweden	Serum IgG	Not investigated	No association ^a^	Secretors higher titers to rotavirus	[23]
Sweden	Neutralization antibody titers	Not investigated	No association	Secretors higher neutralization antibody titers to P[8] but not P[6]	[23]
France	Neutralization antibody titers	Not investigated	Not investigated	Secretors higher neutralization titers to P[8]	[9]
Spain	Salivary IgA	Not investigated	Not investigated	Secretors higher titers to rotavirus	[24]
China	Serum IgG	No association	No association ^b^	Secretors higher titers to VP8* of genotype P[4] and P[8]	[13]

^a^ Lewis status stratified in secretor-positive and negative respectively. ^b^ no Lewis-negative phenotype was observed.

**Table 3 viruses-12-00324-t003:** Association between vaccine take and histo-blood group antigens.

Vaccine	Country	ABO	Lewis	Secretor Status	Method		Reference
					Time Point	Measurement	
**Rotarix**	Nicaragua	B less seroconversion ^a^	Lewis A no seroconversion	Non-secretors less seroconversion ^b^	After 1 dose		[53]
	Pakistan	Non-O less seroconversion compared to O	No association	Non-secretors less seroconversion	After 3 doses		[54]
	Ghana	O less seroconversion compared to B	No association	Non-secretors less seroconversion	After 2-3 doses	Seroconversion	[55]
	Malawi	No association	No association	Non-secretors less seroconversion ^b^	After 2 doses		[15]
**RotaTeq**	Nicaragua	A had most seroconversion ^a^	Lewis A no seroconversion, low power	No association	After 1 dose		[53]
**RV3-BB**	New Zealand	Not investigated	No association, low power	No association	Cumulative		[56]
**Rotarix**	Nicaragua	B no shedding, low power	Lewis A no shedding, low power	Non-secretors no shedding ^b^	After 1 dose		[27]
	Malawi	No association	No association	Non-secretors less shedding	After 1 dose	Vaccine shedding	[15]
	Malawi	No association	No association	No association	After 2 doses		[15]
**RotaTeq**	Nicaragua	No association	Lewis A no shedding, low power	No association	After 1 dose		[27]
**RV3-BB**	New Zealand	Not investigated	No association, low power	No association	Cumulative		[56]

^a^ both non-secretors and secretors. ^b^
*p* > 0.05.

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
