# Peer review of "The Impact of Human Genetic Polymorphisms on Rotavirus Susceptibility, Epidemiology, and Vaccine Take"

_viruses, 2020, doi:10.3390/v12030324_

Round 1

Reviewer 1 Report

Dear authors,

It was a pleasure to read your review "The impact of human genetic polymorphism on Rotavirus susceptibility, epidemiology, and vaccine take".

The only suggestion I have relates to chapter 5 of your review. It seems to me it worth mentioning in the text, not only in the Table 1, that in the study of Lee et.al (ref 13) no P[4] infection was observed in non-secretors. This effect reduced estimates of VE, which therefore was mediated not by reduced susceptibility to P[8] rotavirus infection, but rather by complete protection from P[4] RV. Although only 38 cases of P[4] infection reported in their study, still other authors described even fewer number of samples, and active community surveillance, potentially identifying more mild cases in Bangladesh trial makes these data more valuable

Author Response

Dear authors,

It was a pleasure to read your review "The impact of human genetic polymorphism on Rotavirus susceptibility, epidemiology, and vaccine take".

The only suggestion I have relates to chapter 5 of your review. It seems to me it worth mentioning in the text, not only in the Table 1, that in the study of Lee et.al (ref 13) no P[4] infection was observed in non-secretors. This effect reduced estimates of VE, which therefore was mediated not by reduced susceptibility to P[8] rotavirus infection, but rather by complete protection from P[4] RV. Although only 38 cases of P[4] infection reported in their study, still other authors described even fewer number of samples, and active community surveillance, potentially identifying more mild cases in Bangladesh trial makes these data more valuable

Response: Thank you for your comments. We have now included the suggested information in chapter 5.

Reviewer 2 Report

The manuscript is a well written and provides a clear overview of the area. The role of ABO blood groups and lewis secretor status with regards rotavirus is in its infancy, with many gaps in our knowledge. It is a complicated area with some intriquing data already reported. The review does provide an important update on the data that is available and provides a good attempt at bringing often different and difficult data together, and trying to provide rational explanation for differences in data.

Specific comments:

Line 144 – suggest using individuals rather than people

Line 169 - so does this variation seen in subsequent studies suggest that specific role of lewis/secretor is not as specific as initially suggested, or are studies less well defined?  

Line 193- neut ab are IgG, thus are these levels simply higher in secretors simply because IgG levels are higher. these are not independent factors as suggested.

Line 201- the authors state "whereas in Mesoamerica, the secretor prevalence can reach ~90-95%". Do you mean seropositive or negative?

Line 251- am not sure I follow how pigs support the work. Pigs herds are well defined genetic breeds, selected for growth, meat content etc, and are not simply a group individual pigs. Thus the different data between studies could simply reflect differences in herd, and not really provide supporting information.

Line 260 - most zoonotic transmission events are actually single event transmissions between animal and child and don’t result in spread within population. This is likely simply exposure to high dose.

Line 274 – what is evidence that P6 and p11 have age limits. Infact both cause disease in older infants in India and Africa. The genotype data from these locations tells us this. May wish to clarify – do you mean in settings with a high secretor population, where receptors develop with age. 

Line 306 – I am not sure what the studies where data is provided after 1 vaccine dose actually tells us. We know that both Rotarix and RotaTeq require multiple doses to ensure vaccine take and develop sufficient immunity. It would have been better to compare dose 1, 2 and 3 and see if no difference, and if lack of development does correlate.

Line 322 – id o think at some stage will need to define how studies were undertaken to describe secretor status. For example the study on Rv3-bb was undertaken with sequence analysis whereas the Rix and teq studies were based on saliva antibodies. It is known that variation in ab levels can occur with maternal influence as well as development dependence. Thus it could be likely some of variation in data is due to sampling and testing assays performed.  It maybe useful for authors to comment on this fact, as some data maybe not as reliable as other sets.

Line 331- protection afforded by inability to be infected by rotavirus should in all effect not influence vaccine failure nor the data generated, I imagine those who fail to induce vaccine protection will also be less likely to be infected by wildtype virus, thus in terms of statistical analysis wouldn’t this group have no change, as it would become not be susceptible and wouldn’t become sick.

Figure 1 : is this simplified description and detailed proportions of lewis negatives based on any actual data – do the 1, 4 & 7 actually have and science behind the ratio? Or is just a guess?

Similarly, Fig 2, what data was used to provide evidence that the ratio provided between secretor, non secretor and lewis negative is accurate at all. They are nice simple figures but are used to help your statements, but what evidence is used to justify patterns used.

Table 3, I am unclear what difference is between top and bottom of table, it seem that same info is provided twice.

Author Response

The manuscript is a well written and provides a clear overview of the area. The role of ABO blood groups and lewis secretor status with regards rotavirus is in its infancy, with many gaps in our knowledge. It is a complicated area with some intriquing data already reported. The review does provide an important update on the data that is available and provides a good attempt at bringing often different and difficult data together, and trying to provide rational explanation for differences in data.

Specific comments:

Line 144 – suggest using individuals rather than people

Response: We agree and we have changed “people” to “individuals” throughout the manuscript.

Line 169 - so does this variation seen in subsequent studies suggest that specific role of lewis/secretor is not as specific as initially suggested, or are studies less well defined?  

Response: The association between Lewis-negative phenotype and P[6] infections is strong, both in epidemiological studies and in vitro binding studies. Although discrepancies occur regarding P[4] and P[8], most studies do suggest a secretor specificity. The discrepancies, which we also discuss in the manuscript, can be due to strain variation, study design, methodological differences, or differences in symptoms, as most studies investigate severe cases of rotavirus. Perhaps presence or absence of specific HBGAs is not an essential factor, but a facilitator. We can also view this in Figure 3, as enteroids from non-secretors are infected, but to a lesser extent than secretors. We have further added more comments about methodological differences between studies in paragraph 10. Rotavirus Vaccine Take is Associated with HBGAs

Line 193- Neut ab are IgG, thus are these levels simply higher in secretors simply because IgG levels are higher. these are not independent factors as suggested

Response: The reviewer is correct that there would be an association between neut ab (IgG) and total IgG titer. However as stated in the paragraph, one study observed that “secretors had higher serum rotavirus IgG titers as well as higher neutralization antibody titers to a P[8] strain, but not to a P[6] strain”.  Thus, showing that neutralizing antibodies can be more genotype specific, and we thus decided to include this information.

Line 201- the authors state "whereas in Mesoamerica, the secretor prevalence can reach ~90-95%". Do you mean seropositive or negative?

Response: We refer to secretor positive, and have clarified this in the text.

Line 251- I am not sure I follow how pigs support the work. Pigs herds are well defined genetic breeds, selected for growth, meat content etc, and are not simply a group individual pigs. Thus the different data between studies could simply reflect differences in herd, and not really provide supporting information.

Response: The points we are trying to make here are 1) pigs express similar glycans that are important for susceptibility to rotavirus in humans and 2) there are differences of glycan expression between pigs which could correspond to differences in susceptibility to different rotavirus genotypes, including those who sometimes infect humans. We have now added another statement in the section regarding this to clarify further.

Line 260 - most zoonotic transmission events are actually single event transmissions between animal and child and don’t result in spread within population. This is likely simply exposure to high dose.

Response: We agree with the reviewer. We have now better clarified that HBGA expression is only one of several potential reasons for low human-to-human transmission of the zoonotic strains.

Line 274 – what is evidence that P6 and p11 have age limits. Infact both cause disease in older infants in India and Africa. The genotype data from these locations tells us this. May wish to clarify – do you mean in settings with a high secretor population, where receptors develop with age. 

Response: Although P[11], and particularly P[6] also infects older children, their relative proportion is much higher in neonates compared to other genotypes. Thus, is some settings, they are predominantly found in neonates and not in older children.  We have now added a clarification that this is true for many settings, including more references, but does not have to be true for all settings.

Line 306 – I am not sure what the studies where data is provided after 1 vaccine dose actually tells us. We know that both Rotarix and RotaTeq require multiple doses to ensure vaccine take and develop sufficient immunity. It would have been better to compare dose 1, 2 and 3 and see if no difference, and if lack of development does correlate.

Response: We agree with reviewer that it would be optimal to measure vaccine take after every dose to get the overall picture of vaccine take, which may be difficult due to logistic reasons. However, measuring after the first dose has advantages compared to measurements only after the last dose, as the results are less likely to be biased by natural infections occurring during the vaccination period. We also mention this later in the paragraph “An important factor to consider here is that many studies investigated seroconversion after 2–3 doses, therefore there is a higher likelihood of early natural infections that may affect interpretation of the results.”

Line 322 – id o think at some stage will need to define how studies were undertaken to describe secretor status. For example the study on Rv3-bb was undertaken with sequence analysis whereas the Rix and teq studies were based on saliva antibodies. It is known that variation in ab levels can occur with maternal influence as well as development dependence. Thus it could be likely some of variation in data is due to sampling and testing assays performed.  It maybe useful for authors to comment on this fact, as some data maybe not as reliable as other sets.

Response: This is a good point raised by the reviewer. We have added a comment on methodological differences between studies in the end of the paragraph.

Line 331- protection afforded by inability to be infected by rotavirus should in all effect not influence vaccine failure nor the data generated, I imagine those who fail to induce vaccine protection will also be less likely to be infected by wildtype virus, thus in terms of statistical analysis wouldn’t this group have no change, as it would become not be susceptible and wouldn’t become sick.

Response: We agree with the reviewer on this, as also stated in the paragraph. We do however believe that there are instances when children resistant to the live vaccines, are infected with other rotavirus strains to which they are susceptible. We believe, as we try to argue in the text, that children of the Secretor and Lewis-negative groups, which are low in proportion, might be a group that will 1) not get a good immune response after vaccination and 2) get infected with wildtype P[6] strains.

Figure 1 : is this simplified description and detailed proportions of lewis negatives based on any actual data – do the 1, 4 & 7 actually have and science behind the ratio? Or is just a guess?

Response: It is simplistic as data is lacking for many countries of the continents described, but based on rough estimates of proportions as described from several studies, i.e. 7.5%, 20% and 33%. This has now been clarified in the Figure legends and we have added references. We also slightly modified the Figure for Europe/North America to better describe the rough estimate of ~7.5% Lewis-negatives.

Similarly, Fig 2, what data was used to provide evidence that the ratio provided between secretor, non secretor and lewis negative is accurate at all. They are nice simple figures but are used to help your statements, but what evidence is used to justify patterns used.

Response: They are indeed rough estimates due to the lack of data from many countries, but based on available literature from countries within the continents described in Figure 2. (Approximately 20% non-secretors and 7.5% Lewis-negative in Europe and North America and 20% non-secretors and 33% Lewis-negative in Sub-Saharan Africa. These numbers have now been mentioned in the Figure legend along with references. Also, the Figure has been slightly modified to better account for the stated percepntage of Lewis-negatives within different populations.

Table 3, I am unclear what difference is between top and bottom of table, it seem that same info is provided twice.

Response: The top half of the table is describes seroconversion whereas the bottom of the table describes Vaccine shedding. It was stated in the “Measurement column”. We have now clarified the Table with regards to this.